# Highly Expressed miR-375 is not an Intracellular Oncogene in Merkel Cell Polyomavirus-Associated Merkel Cell Carcinoma

**DOI:** 10.3390/cancers12030529

**Published:** 2020-02-25

**Authors:** Kaiji Fan, Armin Zebisch, Kai Horny, David Schrama, Jürgen C. Becker

**Affiliations:** 1Department of Translational Skin Cancer Research, University Hospital Essen, 45141 Essen, Germany; k.fan@dkfz-heidelberg.de (K.F.); k.horny@dkfz-heidelberg.de (K.H.); 2German Cancer Consortium (DKTK), 45141 Essen, Germany; 3German Cancer Research Center (DKFZ), 69120 Heidelberg, Germany; 4Department of Dermatology, Medical University of Graz, 8010 Graz, Austria; 5Division of Hematology, Medical University of Graz, 8010 Graz, Austria; armin.zebisch@medunigraz.at; 6Otto Loewi Research Center for Vascular Biology, Immunology and Inflammation, Division of Pharmacology, Medical University of Graz, 8010 Graz, Austria; 7Department of Dermatology, University Hospital Würzburg, 97080 Würzburg, Germany; schrama_d@ukw.de; 8Department of Dermatology, University Hospital Essen, 45147 Essen, Germany

**Keywords:** miR-375, antagomiRs, Merkel cell carcinoma, Hippo signaling, focal adhesion

## Abstract

miR-375 is a highly abundant miRNA in Merkel cell carcinoma (MCC). In other cancers, it acts as either a tumor suppressor or oncogene. While free-circulating miR-375 serves as a surrogate marker for tumor burden in patients with advanced MCC, its function within MCC cells has not been established. Nearly complete miR-375 knockdown in MCC cell lines was achieved using antagomiRs via nucleofection. The cell viability, growth characteristics, and morphology were not altered by this knockdown. miR-375 target genes and related signaling pathways were determined using Encyclopedia of RNA Interactomes (ENCORI) revealing Hippo signaling and epithelial to mesenchymal transition (EMT)-related genes likely to be regulated. Therefore, their expression was analyzed by multiplexed qRT-PCR after miR-375 knockdown, demonstrating only a limited change in expression. In summary, highly effective miR-375 knockdown in classical MCC cell lines did not significantly change the cell viability, morphology, or oncogenic signaling pathways. These observations render miR-375 an unlikely intracellular oncogene in MCC cells, thus suggesting that likely functions of miR-375 for the intercellular communication of MCC should be addressed.

## 1. Introduction

Merkel cell carcinoma (MCC) is an aggressive skin cancer. Risk factors for MCC include an advanced age, ultraviolet (UV) light exposure, and immune suppression [1]. About 80% of MCC tumors are associated with genomic integration of the Merkel cell polyomavirus (MCPyV) bearing truncating tumor-specific large T antigen mutations, while the others are characterized by a UV-induced tumor mutational burden [1]. The pathogenesis of these two types of MCC tumors is surmised to be distinct: MCPyV-positive MCC is associated with MCPyV T antigen-mediated tumor suppressor gene inhibition and/or oncogene induction [1,2,3], while in MCPyV-negative MCC tumors, the comparable oncogenic observations are caused by UV-induced DNA mutations [1,4,5,6]. However, the specific molecular alterations caused by either MCPyV or UV-mutations are just starting to emerge [7,8]. 

Transcription factor Atonal homolog 1 is characterized as a lineage-dependency oncogene in MCC, which induces miR-375 expression [9]. microRNAs (miRNAs) are small, ~21nt single-stranded RNAs, which post-transcriptionally regulate the stability and translation of genes, mainly by binding to the 3′ UTR of mRNAs [10,11]. Each miRNA can bind a specific set of genes, which are referred to as its target genes. The dysregulation of miRNAs has been reported in almost all types of human cancer [10,12]. miRNA expression profiling in MCCs revealed miR-375 as one of the most abundant miRNAs in classical MCC cell lines and tumor tissues [13,14,15,16]. Physiologically, miR-375 acts as a pancreatic-islet miRNA essential for β-cell formation and the regulation of insulin secretion [17,18]. Divergent miR-375 expression has been described for multiple cancer types, e.g., reduced expression in gastric [19,20], pancreatic [21], colon [22,23], and liver cancer [24], and high expression in medullary thyroid carcinoma [25], prostate cancer [26], and MCC [13,14,15,16]. Therefore, miR-375 was assumed to be an oncogenic miRNA in the latter group. 

However, when the function of miR-375 in MCC was studied by different groups, the results were inconsistent. Abraham et al. reported that miR-375 was involved in neuroendocrine differentiation and miR-375 knockdown in classical MCC cell lines (MKL-1 and MS-1) and did not alter their growth properties [13]. Our preliminary results of miR-375 knockdown experiments were consistent with their report for the tested MCC cell lines [9]. In contrast, Kumar et al. reported that miR-375 inhibition in WaGa and MKL-1 cells reduced cell growth and induced apoptosis by targeting lactate dehydrogenase b (*LDHB*) [27]. Recent reports from the same group showed that miR-375, together with other miRNAs, inhibits autophagy, thus protecting MCC cells from autophagy-associated cell death [28]. To resolve these controversies, here, we scrutinize the function of miR-375 in MCC. For this, we established a highly efficient method for miR-375 knockdown in classical MCC cell lines and analyzed the inflected effects, with an emphasis on intracellular signaling. 

## 2. Results

### 2.1. Effective Knockdown of miR-375 by Nuclear Transfection Using miR-375 AntagomiRs

To explore the function of miRNAs, it is essential to achieve largely complete knockdown. Achieving highly effective knockdown in classical MCC cell lines is trivial. Therefore, we tested different transfection methods, i.e., lipofectamine and nucleofection, in the two classical MCC cell lines WaGa and PeTa using miR-375 antagomiRs.

The transfection of miR-375 antagomiRs by lipofectamine reduced miR-375 expression in a dose-dependent manner, but was not sufficient for complete knockdown of the highly expressed miR-375 (Figure 1a,b). Next, we performed nucleofection and optimized the transfection conditions. Program D23 with 25nM miR-375 antagomiRs was determined as the optimal protocol for knockdown, which rendered dramatically reduced miR-375 expression in both WaGa and PeTa cells (Figure 1c,d and Appendix A). All further experiments were carried out using these conditions.

### 2.2. miR-375 Knockdown Does Not Impact the Morphology, Proliferative Capacity, or Apoptosis of MCC Cells

We were able to confirm our previous observation that miR-375 knockdown has no major impact on cell proliferation, survival, growth characteristics, or cell morphology (Figure 2 and Appendix A). Notably, even the highly effective miR-375 knockdown did not alter the morphologic appearance as cells still showed a neuroendocrine growth pattern as loose spheroids or single cells, which was identical to the growth pattern in cells transfected with unspecific control antagomiRs (Figure 2a,b). Furthermore, neither the metabolic nor proliferative activity was affected by the miR-375 knockdown (Figure 2c,d). While the harsh transfection conditions for the highly efficient miR-375 knockdown inhibited the proliferation of MCC cells per se, we observed around 40% apoptotic cells 24h after nucleofection in both WaGa and PeTa, and no difference was observed in MCC cells transfected with miR-375 antagomiRs or the negative control (Figure 2e,f and Appendix A). Sequential analyses on days 3 and 5 after transfection further supported that miR-375 knockdown had no specific impact on cell survival or metabolic activity (Figure 2c–f and Appendix A).

### 2.3. miR-375 Target Genes are Involved in Hippo- and EMT-Related Signaling Pathways

To further investigate the role of miR-375 in MCCs, we predicted target genes of this miRNA using the miRNA target prediction tool ENCORI. This tool has the advantage that the results can be filtered for experimentally-validated target genes. Nevertheless, more than 3000 target genes were predicted; thus, the top 500 ranked genes were selected for further analysis (Appendix A). Gene Ontology (GO) analysis showed that miR-375 target genes contribute to several signaling pathways, including Golgi transport, cell junction assembly, Hippo signaling, and neuron differentiation (Figure 3a). To test the relevance of these predictions in MCC, we re-analyzed previously published transcriptome microarray data of MCC cell lines [29]. Of this data set, four MCC cell lines were selected according to their miR-375 expression level: WaGa and MKL-1 with high and, MCC13 and MCC26 with low, miR-375 expression [14]. Gene Set Enrichment Analysis (GSEA) confirmed that particularly genes related to the focal adhesion signaling pathway were lower expressed in cell lines with high miR-375 expression (Figure 3b). Moreover, focal adhesion signaling pathways included most of the experimentally-confirmed miR-375 target genes; this notion also applies for miR-375 target genes related to the Hippo signaling pathway. Both pathways regulate epithelial to mesenchymal transition (EMT) [30] (Figure 3c).

### 2.4. Hippo and EMT Signaling Pathway-Related Genes are Marginally Altered by miR-375 Knockdown

Since our in-silico analysis suggested that miR-375 may regulate Hippo- and EMT-related signaling pathways, we tested this hypothesis by miR-375 knockdown experiments, together with qRT-PCR-based expression arrays for Hippo and EMT signaling-related genes. These experiments, however, did not reveal any statistically significant changes in the gene expression of compounds of these two signaling pathways in MCC cell lines upon miR-375 knockdown. Specifically, miR-375 knockdown only resulted in a non-significant (i.e., less than +/- two-fold change in expression) upregulation of eleven genes (11/84, 13.1%) and downregulation of four genes (4/84, 4.8%) related to the Hippo signaling pathway, as well as a non-significant upregulation of eleven genes (11/84, 13.1%) and downregulation of three genes (3/84, 3.5%) with respect to the EMT-signaling pathway (Figure 4, Appendix A). 

The expression of genes related to Hippo (a) and EMT (b) signaling pathways was determined by a multiplexed qRT-PCR expression array in WaGa cells transfected with anta-375 or anta-NC, normalized to the average Cq values of housekeeping genes (GAPDH, HPRT, and RPLP0) and calculated for the ΔCq of WaGa cells transfected with anta-NC. Gene names colored in red represent genes upregulated upon miR-375 knockdown, while gene names colored in blue represent downregulated genes. Doted lines represent +/- two-fold changes. Experiments were independently repeated twice.

## 3. Discussion

Despite the fact that miR-375 is highly expressed in classical MCC cell lines and MCC tumors, its function in MCC is not clear. To study the relevance of miR-375 in intracellular signaling in detail, we performed a series of knockdown experiments using specific antagomiRs. Surprisingly, even the nearly complete knockdown of miR-375 expression did not affect the proliferation, growth pattern, or cell morphology. Similarly, the impact of miR-375 knockdown on the expression of Hippo and EMT signaling pathway-related genes, i.e., pathways predicted to be regulated by miR-375, was only marginal. These results, taken together with our previous observations that miR-375 is present in MCC cell line-conditioned medium in sera of preclinical xenotransplantation animal models and in sera of MCC patients [14], suggest that miR-375 may serve intercellular rather than intracellular signaling in MCC. Indeed, miR-375 was recently characterized as an exosomal shuttle miRNA [31,32].

In previous reports, miR-375 knockdown or inhibition in MCC cell lines resulted in different consequences. miR-375 knockdown using antagomiRs did not alter growth properties [13], whereas the inhibition of miR-375 using an miRNA sponge suppressed cell growth and induced cell death via downregulation of the *LDHB* gene [27]. Recently, the same group demonstrated that miR-375 inhibits autophagy to protect MCC cells from cell death [28]. AntagomiRs bind particular miRNAs, causing their degradation, while sponge RNAs compete with target mRNAs. Differences in the specificity and/or effectivity of the used methods are likely to explain some of the conflicting results. The quantification of miRNA expression after miR-375 knockdown by antagomiRs might be helpful to better understand this controversy. Notably, miR-375 is lowly expressed in variant MCC cell lines and the ectopic expression of miR-375 decreased their cell viability and migratory potential [13,27], suggesting that miR-375 might be a tumor suppressor in these cells. However, several reports question if these variant MCC cell lines are indeed representative of MCC tumors [29,33].

The knockdown of abundant miRNAs can be challenging [34]. AntagomiRs have been employed for miRNA silencing in vitro and in vivo via miRNA degradation for years [34,35]. In our study, we introduced the respective antagomiRs with two different transfection conditions, which revealed that nuclear transfection was much more efficient and only this method succeeded in nearly complete knockdown up to five days post-transfection. To be noted, we observed a slight increase in miR-375 expression over time after antagomiRs transfection. Therefore, the described method is very effective for short-term knockdown, but not for long-term inhibition (Appendix A). Besides miRNA antagomiRs, the miRNA sponge is another powerful tool that can be employed to inhibit the miRNA function. Notably, an miRNA sponge was used by Kumar et al. to inhibit the miR-375 function in MCC cells [27]. To achieve long-term miRNA inhibition, viral vectors based on stable miRNA antagomiRs or sponge expression and CRISPR-mediated miRNA knockout are feasible [36,37,38]. 

By testing for the expression of Hippo and EMT signaling pathway-related genes after miR-375 knockdown, we observed moderate expression changes of only a few genes. Furthermore, even these changes did not clearly reveal any relevant role of miR-375 in regulating these pathways. Indeed, both EMT-negative (*CAV2* and *IL1RN*) and -positive (*BMP2*, *ITGB1*, and *TGFB2*) regulators were induced upon miR-375 knockdown. Therefore, miR-375 may inhibit or induce EMT in MCC cells. More importantly, none of the changes were greater than two-fold. The gap between the predicted and functional observed effects of miR-375 knockdown is not entirely unexpected. Several reports have provided a possible explanation: long non-coding RNAs, such as *TINCR*, *HNGA1*, and *CircFAT1*, act as an miR-375 sponge [39,40,41]. Alternatively, other miRNAs in MCC may have redundant functional targets as miR-375 [42]. 

In summary, we have demonstrated that even the highly efficient, almost complete knockdown of the highly abundant miR-375 in classical MCC cells lines, has no relevant impact on the cell viability, metabolic activity, morphology, or oncogenic signaling pathways targeted by miR-375. These observations render miR-375 unlikely to function as an intracellular oncogene in MCC cells. 

## 4. Materials and Methods 

### 4.1. Cell Culture 

The classical, MCPyV-positive MCC cell lines WaGa and PeTa were maintained in RPMI-1640 (PAN Biotech, Aidenbach, Germany) supplemented with 10% fetal bovine serum (Sigma-Aldrich, Munich, Germany) and 1% penicillin/streptomycin (Biochrome, Berlin, Germany), as previously described [43]. 

### 4.2. miR-375 Knockdown 

For miR-375 knockdown, specific miR-375 inhibitors (Assay ID: MH10327, Catalog: 4464084, Thermo Fisher Scientific, Frankfurt, Germany) or respective controls (Catalog: 4464076, Thermo Fisher Scientific) were used. 

For the transfection of MCC cells, two methods were compared. Lipofectamine 3000 reagent (Thermo Fisher Scientific) was used according to the instructions of the manufacturer, i.e., 2 × 10^6^ cells were seeded into a 6-well-plate 24 h before transfection with 100 nM or 250 nM of antagomiRs. Alternatively, the Nucleofector™ 2b Device (Lonza, Basel, Switzerland) with the Cell Line Nucleofector^®^ Kit V (Lonza) was used. D-23 was established as the appropriate program to transfect MCC cells (https://bioscience.lonza.com/lonza_bs/CH/en/nucleofector-technology). A total of 100μL of buffer V was mixed with 10 μL miRNA antagomiRs (25 nM) and 2 × 10^6^ MCC cells before being transferred into an electroporation cuvette. After the pulse, cells were immediately transferred into 6-well-plates containing pre-warmed culture media.

### 4.3. qRT-PCR for miR-375

Applied Biosystems TaqMan MicroRNA assays (Thermo Fisher Scientific) were performed according to the manufacturer’s instructions. Pre-designed TaqMan microRNA assays for miR-375 (ID000564) were used. The quantification cycle threshold (Cq) values of miR-375 were normalized to the small nucleolar RNA RNU6B (ID001093) and the relative expression of the respective comparator was calculated using the 2-ΔΔCq method.

### 4.4. MTS Assay

Dead cells and cell debris after nucleofection were removed using Ficoll-mediated gradient centrifugation (Biochrom, Berlin, Germany). For MTS assays, 10^4^ living cells per well of each group (untreated, anta-NC and anta-375) were seeded into 96-well-plates. CellTiter 96 AQ_ueous_ One Solution (Promega, Walldorf, Germany) was used to determine the relative cell proliferation every other day. In brief, 20 uL of the CellTiter solution was added to each well and incubated for two hours, and the absorbance was then measured using a plate reader at 490 nm. 

### 4.5. Apoptosis Assay 

The NucView 488/MitoView 633 apoptosis assay kit (Biotium, Fremont, CA, USA) was used to determine the apoptotic cell rate, according to the manufacturer’s instruction. Viable cells were stained red with MitoView 633 (red, APC-A channel), and apoptotic cells were stained green with NucView 488 (green, PE-A channel). Twenty-four hours’ post-nucleofection and subsequently every other day, cells were analyzed using a CytoFLEX flow cytometer (Beckman Coulter, Krefeld, Germany). 

### 4.6. Prediction of miR-375 Target Genes, Gene Ontology (GO), and Gene Set Enrichment Analysis (GSEA)

The Encyclopedia of RNA Interactomes (ENCORI, http://starbase.sysu.edu.cn/index.php) provides miRNA–target gene interactions, which are based on miRNA target prediction programs, i.e., TargetScan, miRanda, microT, PITA, miRmap, and PicTar. All miRNA target predictions are supported by published Argonaute-crosslinking and immunoprecipitation (AGO-CLIP) data [44]. Predicted target genes are ranked based on the predicted efficacy of targeting, as calculated using cumulative weighted context++ scores of the sites and related AGO-CLIP scores (clipExpNum, Appendix A) [44,45]. The top 500 highest ranking predicted target genes were selected for the following analysis. 

Metascape (http://metascape.org) was applied for GO analysis [46]. Metascape incorporates a core set of default ontologies, including GO processes, KEGG pathways, Reactome gene sets, canonical pathways, and CORUM complexes, for enrichment analysis. 

GSEA, the desktop application from the MSigDB of Broad Institute (Cambridge, MA, USA), was used for re-analysis of the transcriptome microarray of selected MCC cell lines (http://software.broadinstitute.org/gsea/msigdb/index.jsp) [47]. The transcriptome microarray data set GSE50451 was downloaded from Gene Expression Omnibus. Four MCC cell lines were selected and analyzed in GSEA: WaGa and MKL-1 as miR-375_ high, and MCC13 and MCC26 as miR-375_low. 

### 4.7. Pathway Finder Gene Expression Arrays

The RT2 Profiler PCR arrays (SABioscience via Qiagen, Hilden, Germany) for epithelial to mesenchymal transition (EMT) (PAHS-090Z) and Hippo signaling (PAHS-172Z) were performed according to the manufacturer’s instructions. Total RNA was isolated from MCC cell lines three days after nucleofection with miR-375 inhibitors or the respective control. A total of 200ng of total RNA was transcribed into cDNA using the RT2 first strand kit (Qiagen). The relative gene expression was determined using the RT² Profiler PCR Array software from Qiagen (https://dataanalysis.qiagen.com/pcr/arrayanalysis.php). 

### 4.8. Statistical Analysis

Statistical analyses were performed using GraphPad Prism 8.0 Software (GraphPad Software Inc., San Diego, CA, USA). Experiments containing more than two groups were analyzed using the Kruskal–Wallis test, and an unpaired nonparametric ANOVA. R studio (version 3.6.0) was used in the statistical analysis as indicated: the ggpubr R package (version 3.2.0) for the dot plot of gene expression in EMT and Hippo signaling. A *p*-value smaller than 0.05 was considered significant; the respective p-values are indicated in the figures as follows: * *p* < 0.05, ** *p* < 0.01, and *** *p* < 0.001. 

## 5. Conclusions

The highly efficient knockdown of abundant miR-375 achieved by miR-375 antagomiRs with nucleofection did not cause obvious effects on the cell viability, apoptosis, morphology, or oncogenic Hippo- and EMT-related signaling pathways. These observations render miR-375 unlikely to function as an intracellular oncogene in MCC cells. 

## Figures and Tables

**Figure 1 cancers-12-00529-f001:**
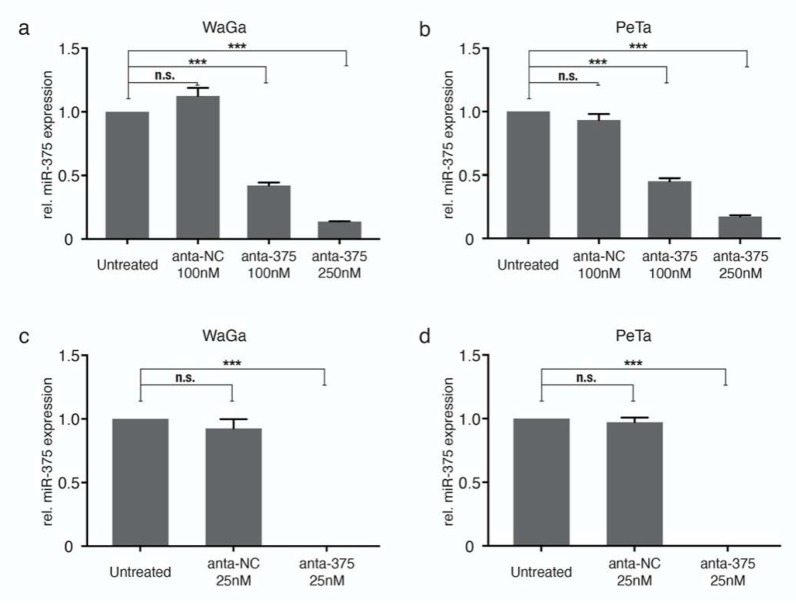
Knockdown of miR-375 in classical Merkel cell carcinoma (MCC) cell lines. Relative miR-375 expression was determined in triplicate by qRT-PCR in WaGa (**a**,**c**) and PeTa (**b**,**d**) cells transfected with miR-375 antagomiRs (anta-375) or a negative control (anta-NC) using lipofectamine (top row; **a**,**b**) or nucleofection (bottom row; **c**,**d**). Quantification cycle threshold (Cq) values were normalized to the small nucleolar RNA RNU6B (U6) and calibrated to the untreated WaGa cells. All experiments were independently repeated three times. Error bars represent SD, *** indicates *p* < 0.001. n.s.: non-significant.

**Figure 2 cancers-12-00529-f002:**
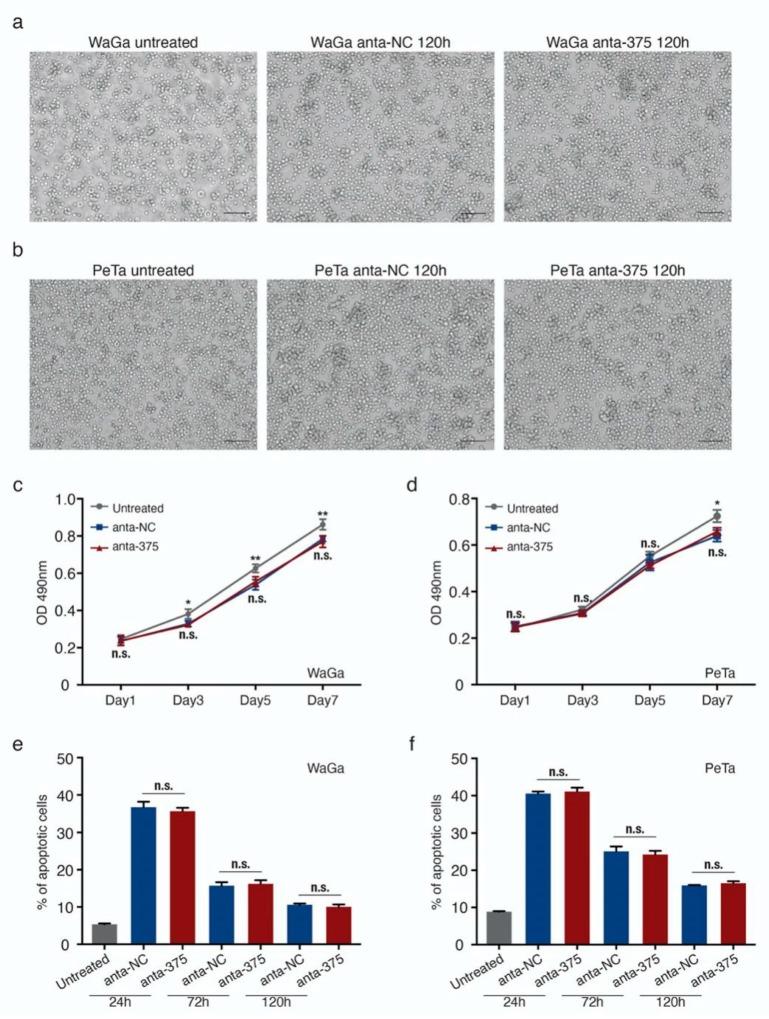
miR-375 knockdown does not alter the cell morphology, viability, and apoptosis of MCC cells. (**a**,**b**) Morphology of WaGa (**a**) and PeTa (**b**) cells, untransfected (untreated) and 120 h after nucleofection with either anta-NC or anta-375. (**c**,**d**) Cell proliferation (metabolic activity) of WaGa (**c**) and PeTa (**d**) cells after nucleofection with miR-375 antagomiRs or a negative control was measured by MTS assays at the indicated time points. Absorbance values at 490 nm are presented. Scale bar: 100 μM. (**e**,**f**) The apoptotic cell rate of untreated or nuclear transfected WaGa (**e**) and PeTa (**f**) cells was determined by flow cytometry using the NucView 488/ MitoView 633 apoptosis assay. Scale bar represents 50µm. All experiments were independently repeated three times, error bars represent SD, * indicates *p* < 0.05, and ** indicates *p* < 0.01. n.s.: non-significant.

**Figure 3 cancers-12-00529-f003:**
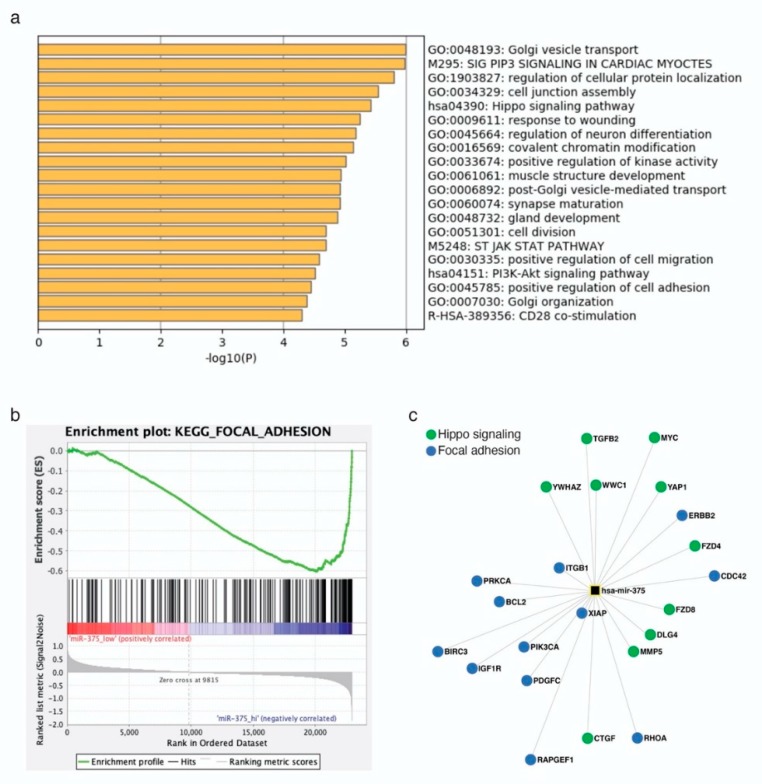
miR-375 target genes are involved in Hippo and epithelial to mesenchymal transition (EMT) signaling pathways in MCC cells. (**a**) Gene ontology analysis was performed in Metascape using the top 500 predicted miR-375 target genes. (**b**) Gene set enrichment analysis was performed using previously published transcriptome microarray data of MCC cell lines with high (WaGa and MKL1) and low (MCC13 and MCC26) miR-375 expression. Enrichment plot of the kegg_focal_adhesion signaling pathway is depicted. (**c**) miR-375 target genes involved in Hippo and focal adhesion signaling pathways.

**Figure 4 cancers-12-00529-f004:**
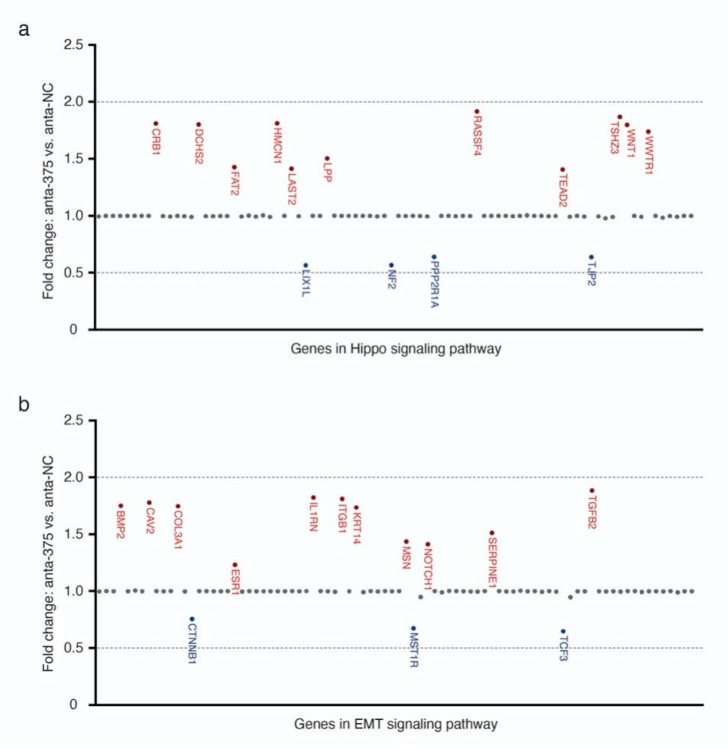
Moderate changes in the expression of Hippo (**a**) and EMT (**b**) signaling pathway-related genes by miR-375 knockdown.

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
