# Peer review of "Highly Expressed miR-375 is not an Intracellular Oncogene in Merkel Cell Polyomavirus-Associated Merkel Cell Carcinoma"

_cancers, 2020, doi:10.3390/cancers12030529_

Round 1

Reviewer 1 Report

The present manuscript investigates the oncogenic potential of miRNA-375 in two Merkel cell carcinoma (MCC) cell lines. The authors managed to knockdown miRNA-375 expression in two virus-positive (VP) MCC cell lines, WaGa and PeTa. This knockdown did not induce alterations in cell proliferation, morphology, viability or the expression levels of target genes of this miRNA in the Hippo and EMT pathways. Thus, in the discussion the authors briefly try to give some explanation to the differences between the predicted and functional effects of miRNA3-75 knockdown.

The authors satisfactorily answered most of the reviewer’s comments following a first round of review. Still, some English revision is needed and minor mistakes need to be corrected (e.g., choose between pRT-PCR and RT-qPCR to use consistently in the text).

Thought the novelty of the article is limited, the authors suggest that miRNA-375 could function extracellularly, opening the door to the study of the role of miRNA-375 in extracellular signaling and exosomes.

Author Response

We wish to express our appreciation for recognizing our revision upon reviewer’s comments. We sincerely apologize for the inconsistent use of certain terms in the manuscript. We have changed the term accordingly, g., qRT-PCR is now consistently used throughout the manuscript.

Reviewer 2 Report

The authors didnot satisfactorily address all the concernes raised. Moreover, A study showing non-significant effect of a miR will not be of considerable effect to readers. A few experiments with an alternative mir should be provied to add some value to the study.

Author Response

Unfortunately, the reviewer did not specify which concerns were not addressed. With respect to relevance, our manuscript addressed the functional role of one of the most abundant miRs in MCC – miR-375. Several previous studies addressed this question, revealing to date controversial results. Here we report, that even the almost complete knockdown of miR-375 in MCC cells had only limited effects on cell proliferation, morphology, viability or the expression levels of target genes of this miRNA in the Hippo and EMT pathways. These observations strongly suggest that miR-375 may rather serve other effects, for example, intercellular signaling. Indeed, its recently described function as exosomal shuttle miR is in line with this notion.

Reviewer 3 Report

The study addresses an interesting topic which falls within the scope of Cancers. Its objective was to investigate in vitro the function of miR-375 in Merkel cell carcinoma cell lines. The authors have highlighted the aims and significance of their work. They have established an efficient method for miR-375 knockdown in MCC cell lines using antagomiRs via nucleofection. However, they were not able to show any significant changes of cell viability, morphology and oncogenic signaling pathways. The conclusions made are supported by the data presented, suggesting that miR-375 is not an intracellular oncogene in MCC cells.

This study is of interest considering the topic addressed. However, even if its results are in line with some previous studies, this paper does not clarify the different results obtained by other groups and deepens the controversy on this subject.

Moreover, the authors have to clarify the different approach in various parts of the manuscript:

For example, in the abstract the authors state that “their expression analyzed by multiplexed RT-qPCR after miR-375 knockdown demonstrating only a limited expression change.”, leaving the impression that there were some changes, even if of small amplitude. But further they state that “high effective miR-375 knockdown in classical MCC cell lines did neither change cell viability, morphology, nor oncogenic signaling pathways”

In the Results section they state that “miR-375 knockdown resulted in an upregulated expression of eleven genes (11/84, 13.1%) and a downregulation of four genes (4/84, 4.8%) related to the Hippo signaling pathway; for EMT-related  pathway related genes,eleven (11/84, 13.1%) were upregulated and three (3/84, 3.5%) downregulated  (Fig. 4).” But further they state that “the changes in gene expression did not reach statistically relevance”

Furthermore, improvements of the language, style and editing are necessary

Author Response

We would like to thank the reviewer for his valuable suggestions. We have now expanded our discussion to include the controversy on this subject (page 7, line 235): Recently, the same group demonstrated that miR-375 inhibits autophagy to protect MCC cells from cell death [28]. AntagomiRs bind particular miRNAs causing its degradation, while sponge RNAs compete with target mRNAs. Differences in the specificity and/or effectivity of the used methods are likely to explain some of the conflicting results. Quantification of miRNA expression after miR-375 knockdown by antagomiRs might be helpful to better understand this controversy We like to thank the reviewer for pointing out these inconsistencies. We have now clarified our statements accordingly: Page 6, line 168: ‘miR-375 knockdown resulted in a moderately upregulated expression of eleven genes (11/84, 13.1%) and a downregulation of four genes (4/84, 4.8%) related to the Hippo signaling pathway. For EMT-signaling pathway related genes, eleven (11/84, 13.1%) were moderately upregulated and three (3/84, 3.5%) slightly downregulated (Fig. 4). However, these changes in gene expression did not reach statistically relevance, i.e., less than +/- two-fold changes in expression. (Fig. 4, Suppl. Table 2).’ Page 6, line 175: ‘Figure 4. Moderate changes in the expression of Hippo and EMT signaling pathway related genes by miR-375 knockdown’ Page 7, line 255: ‘Testing for expression of Hippo and EMT signaling pathway related genes after of miR-375 knockdown, we observed moderate expression changes of only a few genes. Furthermore, even these changes did not clearly reveal any relevant role of miR-375 in regulating these pathways. Indeed, both EMT negative (CAV2 and IL1RN) as well as positive (BMP2, ITGB1 and TGFB2) regulators were induced upon miR-375 knockdown. Thus, miR-375 may inhibit or induce EMT in MCC cells. More importantly, none of the changes were greater than two-fold.

Reviewer 4 Report

The authors have proved that miR-375 knockdown has no impact on MCC cell viability and morphology. At the same time, they have found that the presence of anta-375 had an impact, although minor, on Hippo and EMT signaling related genes. The mentioned signaling pathways are important in the control of skin regeneration.  That is why it would be useful for the Cancers readers dealing with research on MCC and other skin malignancy to know the authors' opinion about the miR-375 role in the regulation of Hippo and EMT signaling. Please supplement the discussion section with more detailed considerations about the miR-375 significance for Hippo and EMT signaling pathways and in connection with this, in the possibility to contribute to cell malignization.

Author Response

We would like to thank the reviewer for his/her valuable insight. We have now improved upon our discussion section to include miR-375’s impact on Hippo and EMT related signaling pathways on page 7, line 257: Testing for expression of Hippo and EMT signaling pathway related genes after of miR-375 knockdown, we observed moderate expression changes of only a few genes. Furthermore, even these changes did not clearly reveal any relevant role of miR-375 in regulating these pathways. Indeed, both EMT negative (CAV2 and IL1RN) as well as positive (BMP2, ITGB1 and TGFB2) regulators were induced upon miR-375 knockdown. Thus, miR-375 may inhibit or induce EMT in MCC cells. More importantly, none of the changes were greater than two-fold.’

Round 2

Reviewer 2 Report

The major issue with the study is that unfortunately there are no significant translational or mechanistic outcome being presented in this study. The authors should validate their findings using a more robust experiment design by atleast using 2 different methods to knockdown miR375  such as  sponge expression, which has been previously shown to be effective in  tumor growth inhibition by miR375 in RCC,  or CRISPR- mediated miRNA knockout should be used to investigate longterm effects.

Author Response

We like to express our gratitude to the reviewer to take the time to read and comment on our manuscript. However, we feel that his comments and suggestions indicate, that he has misinterpreted the aim and the results of our study. The reviewer’s suggestions to use additional approaches for miR-375 inhibition such as knockdown or knockout may be indeed helpful to address the previous controversial results addressing the function of miR-375 in MCC. However, we are not intending to expand our report as a method paper comparing different ways of miRNA inhibition (which would be also unrealistic within the suggested 10 days for revising the manuscript)

Furthermore, we strongly feel that a report demonstrating that a biological process does not exert the assumed canonical functions is valuable, it may prompt analyses of non-canonical functions. The most straight forward hypothesis proposed by other was that miR-375 contributes to the oncogenic phenotype of MCC. However, given the presented results this is apparently not the case. Thus, the message of our report will open new avenues to address the functional role of miR-375 in MCC. Indeed, some earlier observations - which were published in Clinical Cancer Research - suggest that miR-375 may rather serve some intercellular communication. A question we are currently following up.

We addressed this point already in the abstract (page 1, line 27, highlighted): “thus suggesting to address likely functions of miR-375 for intercellular communication of MCC.”

Reviewer 3 Report

The manuscript has been improved. However, the study still doesn't show any significant changes and the authors have not clarified their different scientific approach. In my opinion the correct scientific statement regarding the results of this study would be: " we were not able to show any statisticaly significant changes in gene expression".

If the authors use expressions such as "a moderately upregulated expression of eleven genes and a downregulation of four genes" and "eleven were moderately upregulated and three slightly downregulated" they leave the impression that a difference, even slight or moderate, was shown by this study

Author Response

We thank the reviewer for his thoughtful suggestions. We agree that we should clarify our description of the results of t qRT-PCR based expression arrays for Hippo and EMT signaling related genes. We now specifically state that the observed alterations upon miR-375 knockdown are non-significant. We amended the manuscript as follows (page 6, line 129, highlighted): “Since our in-silico analysis suggested that miR-375 may regulate Hippo and EMT-related signaling pathways, we tested this hypothesis by miR-375 knockdown experiments together with qRT-PCR based expression arrays for Hippo and EMT signaling related genes. These experiments, however, did not reveal any statistically significant changes in gene expression of compounds of these two signaling pathways in MCC cell lines upon miR-375 knockdown. In detail, miR-375 knockdown only resulted in a non-significant (i.e., less than +/- two-fold changes in expression) upregulation of eleven (11/84, 13.1%) and downregulation of four genes (4/84, 4.8%) related to the Hippo signaling pathway, as well as a non-significant upregulation of eleven (11/84, 13.1%) and downregulation of three genes (3/84, 3.5%) with respect to EMT-signaling pathway (Fig. 4, Suppl. Table 2).”

Round 3

Reviewer 3 Report

The manuscript has been improved. 

This manuscript is a resubmission of an earlier submission. The following is a list of the peer review reports and author responses from that submission.

Round 1

Reviewer 1 Report

In the given study authors have investigated the oncogenic potential of miR375 in MCC. miR375 knockdown in 2 different cell lines did not affect cell viability, growth characteristics or morphology. Moreover, miR-375 target oncogenic signaling pathways were unaffected.  

  The authors have shown that knockdown has no significant oncogenic effect in MCC cell lines, the effect of miR375 mimics should be examined to confirm the findings.

The effect on cell viabiliy/ proliferation /metabolic changes should be examined using more rigorous methods including clonogenic survival, orMTS, apoptosis assays

  Minor comment Legends should be provided in Figs 4

Reviewer 2 Report

The role of miRNAs in MCC is a major area of interest that remains largely unexplored in MCC. The present manuscript investigates the oncogenic potential of miRNA-375 in two Merkel cell carcinoma (MCC) cell lines. The authors claim to have optimized the protocol for the complete knockdown of miRNA-375 expression in two virus-positive (VP) MCC cell lines, WaGa and PeTa. This knockdown did not induce alterations in cell fitness. However, in a previous study by the same group, recently published in the Journal of Investigative Dermatology (Fan et al., 2019. MCPyV Large T Antigen-Induced Atonal Homolog 1 Is a Lineage-Dependency Oncogene in Merkel Cell Carcinoma), the results of figure 1 and figure 2 of the present manuscript were shown in supplementary figure S2. The same cell lines were transfected with antagomirs of miRNA-375 by nuclear transfection to knockdown the expression of this miRNA. Similarly, the knockdown had not effect on cell viability. Thus, the only novelty of the present work is the study of the expression of miRNA-375 target genes of the Hippo and EMT pathways, which was not altered upon the miRNA knockdown either.

Regarding the introduction, it seems to be too concise. In the first paragraph, the sentence starting in line 30 should be rephrased to better illustrate the differences between VP and virus-negative (VN) MCCs. At the end of the second paragraph, the authors state that miRNA-375 is overexpressed in MCC. However, this is still controversial since different groups have found reduced expression of this miRNA in VN MCC cell lines and it is argued that miRNA-375 might function as a tumor suppressor in these cells. This is only briefly mentioned by the authors in the introduction and the discussion, arguing that the differences are due to the methods used. Moreover, the authors omit a recently published paper stating a role for miRNA-375, together with other cellular miRNAs, in suppressing autophagy in MCC (Kumar et al., 2019). Since the authors of the present manuscript argued that miRNA-375 is highly expressed in both VP and VN tumors (Fan et al., 2018), it would be interesting to study the effect of miRNA-375 knockdown in relation to the viral status. Besides, the authors suggest that miRNA-375 could function extracellularly, opening the door to the study of the role of miRNA-375 in extracellular signaling and exosomes.

Some specific remarks:

-In the Results, the title of the first subsection may lead to confusion, since the knockdown of the miRNA is achieved by using miRNA antagomirs, which are transfected into the cells.

-In line 56, rather than transfection conditions, transfection methods would be more appropriate.

-In figure 1, anta-NC and anta-375 should be described in the figure legend.

-In each figure, shouldn’t it be included that error bars represent SD of the mean? If that is the case.

-In Suppl. Figure S1, how was the percentage of viable cells determined?

-In Figure 2, should an untrasfected control be included?

Reviewer 3 Report

Fan et al. knocked down miR-375 using antagomiR in two MCPyV-positive MCC cell lines, and showed that inhibition of miR-375 did not have any effect on cell proliferation. Using a bioinformatics approach, the authors functionally annotated the miR-375 targets based on Gene Ontology, and the transcriptome profiles between MCC cell lines with high and low miR-375 levels; which showed that miR-375 target genes are involved in Hippo and EMT signaling pathway. They further observed the effect of miR-375 knockdown on several genes related to Hippo and EMT pathways using RT-qPCR arrays. Overall, the study lacks of appropriate controls and solid experimental support for their conclusion. The manuscript also needs to be improved for clarity, grammatical errors and typos. Specific comments are listed below:

The authors concluded that miR-375 is unlikely an oncogene in MCC. The conclusion was solely based on cell count experiment that revealed no changes in cell number upon miR-375 knockdown. Unfortunately, the authors did not have enough experimental supports for the conclusion, as stated in the title. They should also evaluate other functional effects. On the other hand, the authors claim that miR-375 target genes are involved in Hippo and EMT signaling pathways. Any functional data supporting these pathways would enhance the impact of this study. Why did the authors think that complete knockdown is necessary to study the effect? In my opinion, it is unlikely to achieve complete knockdown using antagomiR or siRNA. The transcription of the gene or miRNA will still occur. Did they look at the functional effect of those lipofectamine-transfected cells? Can the authors also show the U6 levels of the cells transfected with nucleofection? The amplification plot of the transfected cells for both U6 and miR-375 should be presented in the Supplementary material to support that the non-detectable level of miR-375 was not due to reverse transcription and amplification issue. How does antagomiR work? Does it cleave/degrade the miRNA or block the miRNA from reaching its mRNA target? Commonly, miRNA inhibitors bind to specific miRNAs and block their functions, but have no effect on their expression levels. I am wondering whether the chemical modification of antagomiR might interfere reverse transcription of endogenous miR-375, instead of blocking its function. How can the authors demonstrate the antagomiR is functional? Additional experiments, e.g. evaluation of miR-375 known target(s) or reporter assay, would support the inhibition of miR-375 function. Some of the details are missing in the method section, e.g. re-analysis of the transcriptome microarray of the MCC cell lines, cell count assay, number of biological replicates in pathway arrays, etc. On page 6 (line 188-189): Should it be Supplementary Table 1 or 2? The scores are not given in the Table. Numerous grammatical errors are found throughout the manuscript. The manuscript also needs to be improved for clarity and vague statements. Use the same standard throughout the manuscript, e.g. miR-375 (not miRNA-375 or miT375), RT-qPCR (not qRT-PCR), etc. Page 4 (line 106): The subheading does not match the content.

Reviewer 4 Report

In the manuscript, Fan et al describe the optimization of the knockdown of a microRNA using transfection and nucleofection procedures and show successful knockdown of mir-375. They then proceed to show that knockdown does not affect the morphology, proliferation or cell viability of MCC cell lines. They then proceed to undertake a bioinformatic analysis to identify potential target genes that might be regulated by mir-375 using GEA and attempted validation using qRT-PCR. 

Although their protocol for knockdown appears effective and their subsequent GOterm and GEA identifies potential targets, none of the targets reach statistical significant changes. It might be possible that other microRNAs that might be homologous to mir-375 act in a redundant manner and the slight increase in transcript levels might be enhanced by co-transfecting with antagomirs to these paralogs. Have the authors tried to transfect multiple antagomirs? And if so, do they then see viability and other gross defects in the MCC cells ?

It would also be important to identify whether the control RNA in the qRT-PCR (RNA RNU6b) levels themselves are changing in the cells that have survived transfection of the antagomir - as this might indicate something fundamentally changing in these cell lines that might not be apparent in the assays carried out by the authors.

This next comment might be outside of the scope of this study and it is not this reviewer's place to tell the authors how to present their story. However, the methodology used by the authors might be of interest to the community at large. Perhaps using their protocol they might be able to knockdown expression of other microRNAs in difficult to transfect cells? Or perhaps also show that this method works not only for abundant microRNAs but also for those that are present at very low copies in the cell? 

Given my comments, I feel that the methodology developed in the paper has value. However, in its current state, I feel that the manuscript is not suitable for publication